# Risk factors for depression in patients with Parkinson's disease: A nationwide nested case-control study

Yang-Pei Chang[ORCID][1], Min-Sheng Lee[2], Da-Wei Wu[3,4], Jui-Hsiu Tsai[ORCID][5,6]*, Pei-Shan Ho[7,8]*, Chun-Hung Richard Lin[9]*, Hung-Yi Chuang[6,10,11]

1 Department of Neurology, Kaohsiung Municipal Ta-Tung Hospital, Kaohsiung Medical University, Kaohsiung, Taiwan, 2 Department of Pediatrics, Kaohsiung Medical University Hospital, Kaohsiung Medical University, Kaohsiung, Taiwan, 3 Department of Internal Medicine, Kaohsiung Municipal Siaogang Hospital, Kaohsiung Medical University, Kaohsiung, Taiwan, 4 Division of Pulmonary and Critical Care Medicine, Department of Internal Medicine, Kaohsiung Medical University Hospital, Kaohsiung Medical University, Kaohsiung, Taiwan, 5 Department of Psychiatry, Dalin Tzu Chi Hospital, Buddhist Tzu Chi Medical Foundation, Chia-Yi, Taiwan, 6 Program in Environmental and Occupation Medicine, (Taiwan) National Health Research Institutes and Kaohsiung Medical University, Kaohsiung, Taiwan, 7 Department of Oral Hygiene, College of Dental Medicine, Kaohsiung Medical University, Kaohsiung, Taiwan, 8 Division of Medical Statistics and Bioinformatics, Department of Medical Research, Kaohsiung Medical University Hospital, Kaohsiung Medical University, Kaohsiung, Taiwan, 9 Department of Computer Science and Engineering, National Sun Yat-sen University, Kaohsiung, Taiwan, 10 Department of Occupational Medicine, Kaohsiung Medical University Hospital, Kaohsiung Medical University, Kaohsiung, Taiwan, 11 Department of Public Health, Kaohsiung Medical University, Kaohsiung, Taiwan

* faanvangogh@gmail.com (JHT); psho@kmu.edu.tw (PSH); lin@cse.nsysu.edu.tw (CHRL)

**Data Availability Statement:** Data cannot be shared publicly because of the Taiwan National Health Insurance Research Database (NHIRD). Data are available from the NHIRD Institutional

## Abstract

### Objectives

Patients with Parkinson's disease (PD) have higher prevalence of depression than the general population; however, the risk factors for depression in PD remain uncertain.

### Methods/Design

Using the 2000–2010 Taiwan National Health Insurance Research Database, we selected 1767 patients aged ≧ 40 years with new-onset PD during 2000–2009. Among them, 324 patients with a new incidence of depression were enrolled as cases and 972 patients without depression were randomly selected as controls. The groups were frequency-matched at a ratio of 1:3 by age, sex, and index year. Thus, this nested case-control study compared differences between the cases and the controls. Logistic regression models were used to identify risk factors for depression in PD.

### Results

Compared with the controls, the odds ratio (OR) of anxiety disorders in the cases was 1.53 (95% confidence interval [95% CI], 1.16–2.02; P = 0.003), after adjusting for the confounding factors of age, sex, index year, geographic region, urban level, monthly income, and other coexisting medical conditions. The OR for sleep disturbances in the cases was 1.49 (95% CI, 1.14–1.96; P = 0.004) compared to the controls, after adjusting these confounding

Data Access. In this study, there are ethical or legal restrictions on sharing a deidentified data set. Therefore, we have provided contact information for a data access committee, see NHRID_SQL Generator (http://sqlgen.net.nsysu.edu.tw/SQL_Generator/General_Searching.html) in English, cited on 2019/2/13.

**Funding:** No, this study was supported by Grant-in-Hospital-Aid for Financial support by the Kaohsiung Municipal Ta-Tung Hospital, Kaohsiung Medical University, Taiwan (DMR-99-176). The funder had no role in study design, data collection and analysis, decision to publish, or preparation of the manuscript.

**Competing interests:** No authors have competing interests.

factors. Hence, the risk factors for depression in PD were nonsignificantly associated with physical comorbidities.

## Conclusions

In the present study, depression in PD was significantly associated with anxiety disorders and sleep disturbances. Integrated care for early identification and treatment of neuropsychiatric comorbidities is crucial in patients with new-onset PD so as to prevent further PD degeneration.

## Introduction

Parkinson's disease (PD) is the second most common neurodegenerative disease, consisting of both motor and non-motor symptoms. Some non-motor symptoms, including certain cognitive, autonomic, and psychiatric disorders, can precede and even deteriorate the motor symptoms of PD. Depression is one of the most common non-motor symptoms observed in patients with PD, but it is often underestimated in clinical practice [1, 2]. This condition worsens the disability, impairs quality of life, increases the burden on caregivers and society, and may herald dementia with a shortened life expectancy [3–5].

Increasing evidence has suggested that the risk factors for depression in patients with PD are both non-specific (e.g., age, sex, and history of anxiety and/or depression before PD diagnosis) and PD-specific (e.g., the severity of motor symptoms, disease duration, disease stage, the extent of limitations in disease-related activities of daily living, daily levodopa equivalents dose, and the presence of non-motor symptoms such as sleep disturbance, anxiety, hallucinations, and memory-related problems) [6–11]. A previous cross-sectional study showed that non-specific risk factors may be more prominent risk markers of depression in PD than PD-specific factors [6]. Some studies with a longitudinal design and a sample of < 200 patients have reported that a low education level and family history of depression are risk factors for depression in patients with PD [12, 13]. However, these studies have not only been restricted to small-to-medium samples or cross-sectional design, but lacked proper adjustments for potential confounding factors for depression as well.

To determine the risk factors for depression in patients with PD, we conducted a nested population-based cohort study using data derived from the Taiwan National Health Insurance Research Database (NHIRD). The risk factors for depression in PD were determined amongst subjects with depression of varying severity compared to those without depression. Data regarding comorbid anxiety were also included in the analysis.

## Materials and methods

### Data source and ethics

This study used data from the Taiwan NHIIRD, which was developed and is managed by Taiwan National Health Insurance Program (NHIP). Since its inception in March 1995, the NHIP has provided healthcare to approximately 99% of the residents of Taiwan [14]. We accessed the 2000–2010 NHIRD, which contains data on 1 million randomly selected patients (nearly 5% of the total Taiwanese population) drawn in 2010. The NHIRD includes the demographic characteristics, diagnoses, and prescription claims data of each patient. The

prescription claims data contain medication types, prescription dates, medication dosage and the duration of use.

This study was exempt from full review by the Institutional Review Board at Kaohsiung Medical University Hospital (KMUHIRB-EXEMPT(I)-20150043). Informed consent was waived because of the use of previously stored de-identified medical information from the NHIRD.

## Study subjects and design

This nested case-control study [15] used the International Classification of Diseases, Ninth Revision, Clinical Modification (ICD-9-CM) code to identify patients with new-onset PD from January 1, 2000 to December 31, 2009. Study patients were those diagnosed by a neurologist for more than three consecutive visits with first-onset PD (ICD-9-CM code 332.0) who also received anti-PD medication (levodopa, carbidopa, bromocriptine mesylate, pergolide mesylate, amantadine, selegiline, cabergoline, ropinirole, or pramipexole) [16, 17]. The date (year) of first-onset PD was defined as the index date (year). PD patients aged < 40 years on their index date or those who had ever had been diagnosed with dementia, stroke, psychosis, or depression before their index date were excluded. Of all study patients, PD patients with an incidence of depression (ICD-9-CM codes 296.2–296.3, 296.82, 300.4, and 311) diagnosed by a neurologist or psychiatrists after their index date were selected as cases. PD patients without depression were randomly selected as controls after they were frequency-matched to a depressed subject at a ratio of 1:3 according to age, sex, and index year. The flowchart of the nested case-control study is depicted in Fig 1. The ICD-9-CM and Anatomical Therapeutic Chemical (ATC) codes used are provided in S1 Table.

## Covariates

We referred to the diagnosis and prescription files of PD inpatients and outpatients before the index date [18] to ascertain their history of diabetes mellitus, hypertension, chronic pulmonary disease, osteoporosis, chronic heart failure, chronic kidney disease, chronic liver disease, cancer, anxiety disorders, and sleep disturbances by using the ICD-9-CM codes and ATC codes [16, 17, 19]. In addition, sleep disturbances (ICD-9-CM codes 780.5 and 307.4) and hypnotic medications prescribed at bedtime for at least 1 month were the primary focus so as to increase the accuracy of the diagnosis (S1 Table).

## Statistical analyses

We conducted the chi-square test to compare the distribution of sociodemographic characteristics and potential confounding factors between the cases and controls. Logistic regression models were used to analyze the effect of a single and multiple covariates in terms of predicting the risk of depression in PD. All statistical analyses were performed using SAS statistical software (version 9.3, SAS Institute, Cary, NC). A two-tailed $P$-value < 0.05 was considered statistically significant.

## Results

After excluding patients who did not meet the study criteria, 1767 patients with new-onset PD between 2000 and 2009 were selected. Among them, 324 (18.3%) patients with an incidence of depression were enrolled as cases; the mean (stand deviate, SD) time interval between PD and depression occurred was 2.4 (2.3) years. We eventually matched 324 cases with 972 controls whose new-onset PD patients without depression were randomly selected as controls after

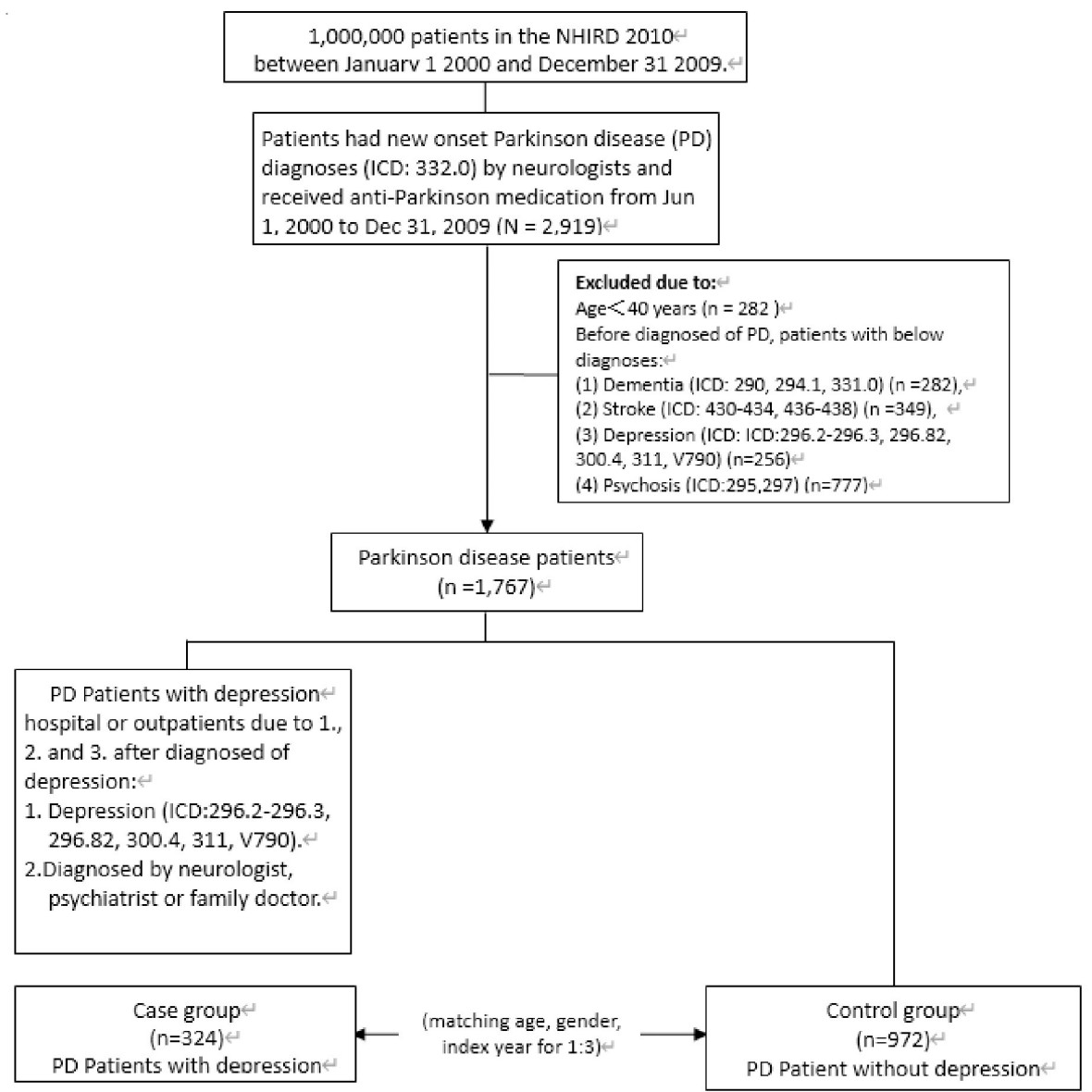

**Fig 1. Flowchart of the nested case-control study from prescriptions in the 2000–2010 Taiwan National Health Insurance Research Database.**
Abbreviations: PD, Parkinson Disease; NIHRD, National Health Insurance Research Database; ICD, International Classification of Diseases.

they were frequency-matched with cases at a ratio of 1:3 based on age, sex, and index year (Fig 1). In our study, the mean (SD) follow-up times after PD was 4.5 (3.0) years. Table 1 shows the distribution of sociodemographic characteristics and coexisting medical conditions of the cases and controls. A significantly higher percentages of cases had anxiety disorders (36.1% vs. 28.2%, $P = 0.007$) and sleep disturbances (38.9% vs. 31.0%, $P = 0.009$) compared with controls. The differences between cases and controls were nonsignificant for the other coexisting medical conditions and all sociodemographic characteristics.

Table 2 lists the adjusted odds ratios (adjusted ORs) for depression in the study patients. After adjusting for age, sex, index year, geographic region, urban level, monthly income, emergency visits before PD diagnosis, PD with dementia, the Charlson Comorbidity Index score, and other comorbidities, anxiety disorders and sleep disturbances were the only two factors

**Table 1. Characteristics of Parkinson's disease patients with depression (cases) and the controls.**

| Characteristics | The cases[a] PD with depression (n = 324) | | The controls[a] PD without depression (n = 972) | | p value |
|---|---|---|---|---|---|
| | n | (%) | n | (%) | |
| **Sociodemographic characteristics** | | | | | |
| **Sex** | | | | | |
| Male | 148 | 45.7 | 444 | 45.7 | 1.000 |
| Female | 176 | 54.3 | 528 | 54.3 | |
| **Age (years)** | | | | | |
| 40–49 | 13 | 4.0 | 39 | 4.0 | 1.000 |
| 50–59 | 35 | 10.8 | 105 | 10.8 | |
| 60–69 | 86 | 26.5 | 258 | 26.5 | |
| > 70 | 190 | 58.6 | 570 | 58.6 | |
| **Geographic region** | | | | | |
| Northern | 134 | 41.4 | 432 | 44.4 | 0.616 |
| Central | 85 | 26.2 | 248 | 25.5 | |
| Southern and Eastern | 105 | 32.4 | 292 | 30.1 | |
| **Urban level** | | | | | |
| Urban and suburban | 232 | 71.6 | 673 | 62.2 | 0.422 |
| Rural | 92 | 28.4 | 299 | 30.8 | |
| **Monthly income (NT$)** | | | | | |
| High (>30,000) | 28 | 8.6 | 116 | 11.9 | 0.102 |
| Low (<30,000) | 296 | 91.4 | 856 | 88.1 | |
| **Coexisting medical conditions** | | | | | |
| **Emergency[b]** | | | | | |
| No | 213 | 65.7 | 682 | 70.2 | 0.136 |
| Yes | 111 | 34.3 | 290 | 29.8 | |
| **PD with dementia[c]** | 44 | 13.6 | 168 | 17.3 | 0.119 |
| **CCI score** | | | | | |
| 0 | 50 | 15.4 | 141 | 14.5 | 0.363 |
| 1–2 | 124 | 38.3 | 337 | 34.7 | |
| >2 | 150 | 46.3 | 492 | 50.8 | |
| **Comorbidities** | | | | | |
| Diabetes mellitus | 109 | 33.6 | 334 | 34.4 | 0.813 |
| Hypertension | 202 | 62.3 | 575 | 59.2 | 0.310 |
| Chronic pulmonary disease | 147 | 45.4 | 482 | 49.6 | 0.188 |
| Osteoporosis | 93 | 28.7 | 272 | 28.0 | 0.803 |
| Chronic heart failure | 3 | 0.9 | 7 | 0.7 | 0.718 |
| Chronic kidney disease | 64 | 19.8 | 172 | 17.7 | 0.406 |
| Chronic liver disease | 103 | 31.8 | 301 | 31.0 | 0.782 |
| Cancer | 8 | 2.5 | 43 | 4.4 | 0.117 |
| Anxiety disorders | 117 | 36.1 | 274 | 28.2 | 0.007 |
| Sleep disturbance | 126 | 38.9 | 301 | 31.0 | 0.009 |

Abbreviations: CCI, Charlson Comorbidity Index; NT$, new Taiwan dollars; PD, Parkinson disease.

[a]Values are numbers (percentage) of column totals of patients.

[b]Visiting the emergency before diagnosis of PD.

[c]Dementia occurred at least one year after diagnosis of PD.

**Table 2. Risk factors for depression in the study patients with Parkinson's disease (n = 1296).**

| Variables | Adjusted OR[a] | 95% CI[a] | p value |
|---|---|---|---|
| **Comorbidities** | | | |
| Diabetes mellitus | 1.05 | (0.78–1.42) | 0.743 |
| Hypertension | 1.21 | (0.91–1.59) | 0.188 |
| Chronic pulmonary disease | 0.88 | (0.66–1.17) | 0.375 |
| Osteoporosis | 1.05 | (0.77–1.42) | 0.753 |
| Chronic heart failure | 1.29 | (0.33–5.02) | 0.719 |
| Chronic kidney disease | 1.24 | (0.88–1.75) | 0.218 |
| Chronic liver disease | 1.10 | (0.83–1.46) | 0.508 |
| Cancer | 0.56 | (0.27–1.28) | 0.180 |
| Anxiety disorders | 1.53 | (1.16–2.02) | **0.003** |
| Sleep disturbance | 1.49 | (1.14–1.96) | **0.004** |

Abbreviations: CCI, Charlson Comorbidity Index; CI, confidence interval; OR, odds ratio; PD, Parkinson's disease.
[a]Adjusted for age, sex, the index year, geographic region, urban level, monthly income, emergency visits before PD diagnosis, PD with dementia, PD with psychosis, CCI score, and other comorbidities.

that significantly affected depression in patients with PD, with an adjusted OR of 1.53 (95% confidence interval [95% CI], 1.16–2.02; $P = 0.003$) for anxiety disorders and 1.49 (95% CI, 1.14–1.96; $P = 0.004$) for sleep disturbances. The association with depression in patients with PD was non-significant for all physical comorbidities, including diabetes, hypertension, chronic pulmonary disease, osteoporosis, chronic heart failure, chronic kidney disease, chronic liver disease, and cancer (all $P > 0.05$).

## Discussion

We conducted a nationwide nested case-control study to determine the risk factors for depression in patients with PD. There were 1767 patients with new-onset PD identified from 2000 to 2009 in this analysis. Overall, 324 (18.3%) patients with PD had an incidence of depression; the mean (SD) time to depression occurrence was 2.4 (2.3) years. The results revealed that the comorbidities of anxiety disorders and sleep disturbances were risk factors for depression in patients with PD, independent of physical comorbidities.

In the present study, the incidence of depression in Taiwanese patients with PD was 18.3%, as assessed from clinical visit records. The result corresponds with the incidence found in a previous 5-year longitudinal prospective cohort study, which showed that approximately 20% of PD patients suffer from depression [20]. Hence, our incidence seems to be below the average incidence value (about 35%) of international reports [21]. Apart from differences in the methods used to assess depression, differences in study populations, and varying statistical measures [13, 21], this lower incidence is most likely due to the lack of awareness of depression in PD of clinicians, the patients, and their families. The overlapping symptoms of PD and depression—such as psychomotor retardation, facial expression, fatigue, insomnia, and poor food intake—may make it difficult to the diagnose depression in patients with PD [22, 23]. Primary care physicians, particularly neurologists treating movement disorders and psychiatrists in Taiwan, should consider developing consensus on the diagnosis of depression in PD patients.

In this study, we analyzed the associated between depression in PD and its comorbidities. Four previous studies with a cross-sectional design and sample sizes < 250 patients with PD have analyzed this association [6, 8, 11, 12]. Wichowicz et al [11] reported that typical risk factors for depression in patients with PD are strongly associated with a family history of

depression and weakly associated with somatic comorbidities; however, these researchers did not assess anxiety and sleep disturbances [12]. Another Polish population-based study concluded that sleep disturbances and the disease severity of PD were the two risk factors significantly associated with depression in PD [11]. In addition, two other studies found that comorbidities, including anxiety, a family history of depression, sleep disturbances, memory-related problems, hallucinations, and postural hypotension were more commonly identified in PD patients with depression than in those without depression, whereas the incidence of physical disorders such as head injury and hypertension differed nonsignificantly between these two types of patients [6, 8]. The results of the present study were consistent with those of previous research, which found that depression in PD is strongly associated with the neuropsychiatric comorbidities of anxiety disorders and sleep disturbances, but nonsignificantly associated with physical comorbidities. These neuropsychiatric comorbidities are among the non PD-specific risk factors for depression in PD, which were associated with a three-fold greater risk of depression than PD-specific factors in PD [6]. Early prediction of risk factors for depression in PD is key to preventing further PD degeneration. However, future studies must determine the extent to which treatment for such neuropsychiatric comorbidities effectively prevents subsequent PD degeneration.

The etiology of depression in PD is most likely multifactorial, with a possible contribution from both psychosocial stress and brain-related changes [24, 25, 26]. At the time of initial PD diagnosis, patients and their families must make adjustments to address this chronic progressive and debilitating brain disorder, which may lead to loss of jobs, marital discord, increasing withdrawal, and a worsening quality of life. Some patients, particularly those with early-onset PD, exhibit certain aspects of a reactive emotion such as sadness, anxiety, and sleep disturbances. Increasing evidence has recently implied that depression in PD is secondary to brain-related changes, and is a reaction to psychosocial stress and the associated disability [24]. Brain-related changes may result from the severity and course of PD, the adverse effects of medications, or the comorbidities of the disorder [1, 24, 25]. Furthermore, we found that these brain-related changes were strongly associated with neuropsychiatric comorbidities, namely anxiety disorders and sleep disturbances, and weakly associated with physical comorbidities. Anxiety is a common non-motor symptom in PD, with a reported prevalence of 25% to 49% [27, 28]. Moreover, 14% to 40% of PD patients are diagnosed with anxiety comorbid with depression [27, 29]. We found a similar prevalence of anxiety disorders in PD patients with depression (36%) and in those without depression (28.2%). Growing evidence has shown that dopaminergic treatment has the potential not only to improve depression in patients with PD [28], but also to induce neuropsychiatric disorders (e.g., mania, psychosis, impulsive control disorder, and dopamine dysregulation syndrome) [30], implying that dopaminergic neuron loss is likely to occur in PD patients with depression and anxiety. Extra-striatal and non-dopaminergic neurotransmitter systems have become increasingly recognized for their role in the course of PD [31]. Serotonergic raphe nuclei and the noradrenergic locus coeruleus implicated in depression have a disposition to lewy body [32, 33], which precedes basal midbrain involvement in PD [31]. Both anxiety and depression have been observed to induce similar changes in norepinephrine and serotonin systems; anxiety disorders may occur earlier than depression, and even before motor symptoms appear in PD. This progression may explain the occurrence of depression and anxiety in PD during the premotor phase [34]. Elevated inflammatory cytokine, soluble interleukin 2 receptor (sIL-2R) and tumor necrosis factor-α, may be associated with depression and anxiety in PD, which also imply the underlying neuro-inflammatory process may contribute to the development of depression and anxiety in PD [35]. This study found anxiety to be a crucial risk factor for depression in PD; this finding is consistent with those reported in previous anatomical, pathological, and pharmacological studies.

Sleep disturbances are among the most common non-motor symptoms of PD [36]. Sleep disturbances such as sleep fragmentations and early awakenings can lead to poor sleep quality and excessive daytime sleepiness, which may exacerbate motor performance and degrade the quality of life of patients with PD [37, 38]. A previous study found strong genotypic correlations between sleep disorders and depression in PD; however, the causality remains unknown [39]. The present study can only confirm the association between sleep disturbance and depression in PD.

## Strength and limitation in the study

The strengths of our study were the use of nationwide population-base data and the large sample. We analyzed the risk factors for depression in patients with new-onset PD. The results are relevant for PD patients with comorbid depression, particularly in the ethnic Chinese population. To increase the accuracy of PD diagnosis, an ambulatory care expenditure database (containing diagnoses by neurologists using ICD-9-CM codes) and a prescription claims database (containing PD treatment medications) were used to confirm the PD diagnosis. Furthermore, covariates, including underlying common physical and psychiatric disorders, were considered. Despite its retrospective nature, we used a nested case-control design to analyze the national population-based database, and avoided two major potential biases, namely selection bias and recall bias.

Our study has certain limitations, and hence, could not control for all potential biases. First, the prevalence of PD, depression, and comorbidities, such as sleep disturbances, may have been underestimated in medical claims, because certain patients experiencing mild symptoms of such disorders are unlikely to visit medical services. Consecutive outpatient visits and medications by neurologists may warrant symptomatic effects of dopaminergic treatment but could exclude the possibility of partial response in atypical parkinsonism patients such as multiple system atrophy or progressive supranuclear palsy. Moreover, we could not assess the polysomnography data to confirm the diagnosis of sleep disorders such as rapid eye movement disorder, restless leg syndrome, and obstructive sleep apnoea, symptoms typically observed in patients with PD. Second, information on other risk factors contributing to depression in PD, such as biochemistry data, current alcohol consumption, head injury and the severity of PD and other comorbidities such as apathy executive dysfunction or mild cognitive impairment, were unavailable in the database. Alcohol consumption is widely known for its positive association with depression. We were unable to explain the positive association observed in the study because of the lack of data. Other lifestyle-related risk factors for depression, including coffee consumption, exposure to herbicides, and a childhood history of traumas such as child abuse, were not included in our study. Finally, various aspects of depression in PD were also unavailable in the study, because we used the ICD-9-CM Diagnosis Coding System for depression diagnosis. Further, we can consider the following two systems as a diagnostic nomenclature of depression in PD: the Diagnostic and Statistical of Mental Disorders (5th Edition) by American Psychiatric Association and the provisional diagnostic criteria for depression in PD proposed by the NINDS/NIMH Work Group [40].

## Conclusions

The findings of the present study demonstrate that the risk factors for depression in patients with PD are significantly associated with the comorbidities of anxiety disorders and sleep disturbances, but nonsignificantly associated with physical comorbidities. Based on our findings and previously reported evidence, physicians should emphasize the early identification and treatment of neuropsychiatric comorbidities in patients with new-onset PD so as to prevent further PD degeneration.

## Supporting information

**S1 Table. ICD-9-CM codes and ATC codes used in this study.**
(DOC)

## Acknowledgments

We thank the Taiwan NHRI and BNHI for providing data. The interpretation and conclusions contained in this article do not represent those of the NHRI and BNHI. We would like to thank Wallace (www.editing.tw) for English-language editing.

## Author Contributions

**Conceptualization:** Yang-Pei Chang, Jui-Hsiu Tsai, Pei-Shan Ho.

**Data curation:** Yang-Pei Chang, Min-Sheng Lee, Da-Wei Wu.

**Formal analysis:** Pei-Shan Ho, Chun-Hung Richard Lin.

**Funding acquisition:** Min-Sheng Lee.

**Methodology:** Jui-Hsiu Tsai, Pei-Shan Ho, Chun-Hung Richard Lin.

**Project administration:** Yang-Pei Chang, Min-Sheng Lee, Da-Wei Wu, Jui-Hsiu Tsai.

**Software:** Chun-Hung Richard Lin.

**Supervision:** Pei-Shan Ho, Chun-Hung Richard Lin, Hung-Yi Chuang.

**Writing – original draft:** Yang-Pei Chang, Min-Sheng Lee, Da-Wei Wu, Jui-Hsiu Tsai.

**Writing – review & editing:** Jui-Hsiu Tsai, Pei-Shan Ho, Chun-Hung Richard Lin, Hung-Yi Chuang.

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
