## [Decision Letter · Decision Letter 0]

27 May 2020

PONE-D-20-09123

Risk factors for depression in patients with Parkinson’s disease: A nationwide nested case-control study

PLOS ONE

Dear Dr. Tsai,

Thank you for submitting your manuscript to PLOS ONE. After careful consideration, we feel that it has merit but does not fully meet PLOS ONE’s publication criteria as it currently stands. Therefore, we invite you to submit a revised version of the manuscript that addresses the points raised during the review process.

We look forward to receiving your revised manuscript.

Kind regards,

Mathias Toft, MD, PhD

Academic Editor

PLOS ONE

Journal Requirements:

2. Please include your tables as part of your main manuscript and remove the individual files. Please note that supplementary tables should remain as separate "supporting information" files.

Reviewers' comments:

Reviewer's Responses to Questions

**Comments to the Author**

1. Is the manuscript technically sound, and do the data support the conclusions?

Reviewer #1: Yes

Reviewer #2: Yes

Reviewer #3: Yes

2. Has the statistical analysis been performed appropriately and rigorously? 

Reviewer #1: Yes

Reviewer #2: Yes

Reviewer #3: Yes

3. Have the authors made all data underlying the findings in their manuscript fully available?

Reviewer #1: No

Reviewer #2: Yes

Reviewer #3: Yes

4. Is the manuscript presented in an intelligible fashion and written in standard English?

Reviewer #1: Yes

Reviewer #2: Yes

Reviewer #3: Yes

5. Review Comments to the Author

Reviewer #1: Thank you for the opportunity to comment on this article describing a nested case-control study of depression in Taiwanese Parkinson's disease patients. The article is clearly written and the methods are well-described. The conclusions are that anxiety and sleep disorders are more common in PD cases who have co-morbid depression compared to those who do not. This finding aligns with other reports; the results are unremarkable.

There are a few limitations of the study that should be further discussed by the authors:

1. An exclusion criteria for the PD with depression group is the diagnosis of depression that precedes PD presentation. To my mind this is problematic as it is well-accepted that many PD patients report a history of depressive symptoms many years before the presentation of motor Parkinsonism. This criteria may significantly lead to an under-estimation of the frequency of depression in this disease group. Can the authors comment?

2. The inclusion of prescriptions of dopaminergic Parkinson medication as a means of refining diagnosis is understandable - was there a need for a symptomatic response to the medication in the inclusion criteria?

3. It is very possible that symptoms of depression (that may warrant a DSM diagnosis of depression) may not have been screened for in the patients in the PD without depression group. While the authors touch on this in the discussion, it is very important to note that absence of evidence is not evidence of absence. How did the authors address this situation?

Further, despite formal diagnosis, some PD cases are prescribed anti-anxiolytics and anti-depressive medications "to help them sleep or relax". Was there an attempt to look at medication usage in this group to confirm that they were not on such medications?

4. Both anxiety and sleep disturbance (particularly REM behavioural sleep disorder) are recognised parts of the spectrum of PD symptoms. Can the authors comment on when these were recognised in the PD cases with respect to PD diagnoses? Did the study also consider other diagnoses prior to the incident diagnosis of PD (eg head injury was one that would be interesting to consider).

Finally, was there any investigation of disease severity (by any measure) or Quality of Life for patients in either group? Such information would highlight the importance of recognising and treating depression in PD patients.

The dataset accessed as part of this report is a valuable one and further analyses are likely to provide important insights into Parkinson's disease.

Reviewer #2: Next, some observations:

1. There are several symptoms that share apathy and depression, how did the researchers control this confounding factor?

2. Up to 20% of patients with Parkinson's disease de novo may have mild cognitive impairment. In that sense, it should be clear that these patients were excluded.

3. Patients with Parkinson's disease and subjective cognitive complaints may have executive dysfunction, which could increase suspicion of depression/apathy. How did the researchers control for this confounding factor?

4. It is important to comment on the likelihood of patients with low levels of education, as in that case, cognitive tests should report on their adaptation and validation.

Reviewer #3: Manuscript Number: PONE-D-20-09123.

Title: Risk factors for depression in patients with Parkinson’s disease: A nationwide nested case-control study.

The authors report a nested case-control study for risk factors of depression in Parkinson’s disease among 324 patients with depression and 972 patients without depression, using the 2000–2010 Taiwan National Health Insurance Research Database. They found an odds ratio of 1.53 (95% confidence interval, 1.16–2.02; P = 0.003) for anxiety disorders after adjusting for the confounding factors of age, sex, index year, geographic region, urban level, monthly income, and other coexisting medical conditions. For sleep disturbances, the odds ratio was 1.49 (95% confidence interval, 1.14–1.96; P = 0.004) compared to the controls, after adjusting these confounding factors. I enjoyed reading this interesting manuscript. I have only a few suggestions.

Major points.

1. Various kinds of depression.

Authors presented ICD codes for various kinds of depression. However, these codes could not be so apparent for readers to describe various aspects of depression in Parkinson’s disease. We can consider the following two systems as a diagnostic nomenclature of depression in Parkinson’s disease.

1-1. One, the Diagnostic and Statistical Manual of Mental Disorders (5th Edition) by American Psychiatric Association classifies depression according to its etiology and symptom severity. This system comprises of major depression, minor depression, dysthymia, adjustment disorder, and depressive disorder due to Parkinson’s disease.

1-2. The other, the provisional diagnostic criteria for depression in Parkinson's disease proposed by the NINDS/NIMH Work Group classifies depression according to the presence or absence of symptoms and severity. This system includes major depression, minor depression, dysthymia, and subsyndromal depression.

1-3. I would like to ask authors for supplementing these systems in addition to the ICD codes. If not possible, I’d like to suggest that this aspect could be pointed out as a limitation of this study.

2. Risk factors versus association factors.

If anxiety disorder and sleep disorder are risk factors to develop depression, we can observe a temporal order of preceding risk factors and resulting depression. If not, anxiety disorder and sleep disorder could be ascribed as association factors. Please clarify this point.

Minor points.

3. Page 35. Figure 1.

In the box of “Excluded due to”, ICD 300.3.

In the box of “PD patients with an incidence of”, ICD 300.4.

Please check 300.3.

6. PLOS authors have the option to publish the peer review history of their article (what does this mean?). If published, this will include your full peer review and any attached files.

Reviewer #1: Yes: George D. Mellick

Reviewer #2: Yes: Nilton Custodio

Reviewer #3: No

---

## [Author Response · Author response to Decision Letter 0]

11 Jun 2020

Reviewer #1: Thank you for the opportunity to comment on this article describing a nested case-control study of depression in Taiwanese Parkinson's disease patients. The article is clearly written and the methods are well-described. The conclusions are that anxiety and sleep disorders are more common in PD cases who have co-morbid depression compared to those who do not. This finding aligns with other reports; the results are unremarkable. 

There are a few limitations of the study that should be further discussed by the authors:

 1. An exclusion criteria for the PD with depression group is the diagnosis of depression that precedes PD presentation. To my mind this is problematic as it is well-accepted that many PD patients report a history of depressive symptoms many years before the presentation of motor Parkinsonism. This criteria may significantly lead to an under- estimation of the frequency of depression in this disease group. Can the authors comment?

ANS: Thanks for your comments.

 As depression may be a risk factor and a common prodromal symptom of PD patients, it is possible for our study to under-estimate our PD patients with coincided depression, which may be another explanation for the relatively low incidence of depression in our PD patients. Because of the convenience of our insurance system, in Taiwan citizens who suffered from depression can be treated earlier than those in other countries. In the same time, many clinicians have also observed many patients are diagnosed with drug-induced parkinsonism due to doctoral shopping for treating their symptoms. To clarify the temporality of association in the risk factors for depression in patients with PD, we decided to exclude patients with past history of depression in order to minimize the inclusion of patients with drug-induced parkinsonism, which may complicate the interpretations of the results.

 2. The inclusion of prescriptions of dopaminergic Parkinson medication as a means of refining diagnosis is understandable - was there a need for a symptomatic response to the medication in the inclusion criteria?

ANS: Thank you for this comment.

 To ensure the symptomatic response to the medications are sustainable for defining our diagnosis, we include patients with more than 3 consecutive outpatient visits with medications by neurologists. However, it is still possible that we were not be able to completely exclude patients with multiple system atrophy, progressive supranuclear palsy or other atypical parkinsonism. We would revise our article and point out the limitations. See them in detail on the Line 12 of Page 8 in the Materials and Methods section and on the Lines 7-10 of Page 17 in the “Strength and limitation in the study” of the Discussion section.

 3. It is very possible that symptoms of depression (that may warrant a DSM diagnosis of depression) may not have been screened for in the patients in the PD without depression group. While the authors touch on this in the discussion, it is very important to note that absence of evidence is not evidence of absence. How did the authors address this situation? Further, despite formal diagnosis, some PD cases are prescribed anti-anxiolytics and anti-depressive medications "to help them sleep or relax". Was there an attempt to look at medication usage in this group to confirm that they were not on such medications?

ANS: 

 As many PD patients with depression may also respond to dopaminergic medications like dopamine agonists, PD with subclinical depression may not be screened by the clinicians because of the pharmacological effects. We hope the article may contribute to the increasing attention on the issue and the development of the consensus of screening depressive PD patients with validated scale and correct diagnosis.

 Further, we believed that this comment often happened in the clinic. In our health insurance system, it is required that the consistency of prescription medications and diagnosis code. In our study, we have excluded dementia, stroke, psychosis, and depression before PD diagnosis. So the anti-anxiolytics and anti-depressive medications could be ignored.

 4. Both anxiety and sleep disturbance (particularly REM behavioural sleep disorder) are recognised parts of the spectrum of PD symptoms. Can the authors comment on when these were recognised in the PD cases with respect to PD diagnoses? Did the study also consider other diagnoses prior to the incident diagnosis of PD (eg head injury was one that would be interesting to consider).

ANS: Thanks for your comments.

 Anxiety and sleep disturbances were diagnosed before the index date of PD diagnosis and depression diagnosis (in depressive PD group) using both ICD codes and ATC codes in order to confirm the diagnosis and the diagnosis should be made by neurologists or psychiatrists with regular outpatient follow-ups. We did not put head injury prior to PD diagnosis to investigate the association but it would be a very interesting topics to be investigated.

Finally, was there any investigation of disease severity (by any measure) or Quality of Life for patients in either group? Such information would highlight the importance of recognising and treating depression in PD patients.

ANS: Thanks for your comment.

 No, we did not assess and confirm quality of life for our patients because all the data had been delinked to avoid personal identity as we mentioned in the limitations.

 But considering disease severity in terms of the initial dosage of dopaminergic medications, mean Levodopa equivalent daily dose of depressive and non-depressive groups were 116.26 ± 32.35 mg and 143.81 ± 59.34 mg respectively without significant difference. (data not shown) In addition, comorbidity score or emergency visits before PD diagnosis did not differ significantly between two groups. 

The dataset accessed as part of this report is a valuable one and further analyses are likely to provide important insights into Parkinson's disease. 

Reviewer #2: Next, some observations: 

 1. There are several symptoms that share apathy and depression, how did the researchers control this confounding factor?

ANS: Thank you for important comments.

Like depression, apathy is also a common non-motor symptom in Parkinson’s disease (PD), but is often under-recognized. Apathy is defined as a lack of motivation characterized by reduced emotional expression and diminished goal-oriented behavior. Apathy may share similar feature with depression. Apathetic PD patients may represent as decreased self-generated activities as a consequence of decreased rewarding experiences, but depressive PD patients describe the increase or enhancement of sadness, feelings of guilt, recurrent (and even involuntary intrusion of) negative thoughts and/or feelings, helplessness, hopelessness, pessimism, self-criticism, anxiety, and even suicidal ideation. Differentiation of apathy from depression would be difficult and may be mainly based on a well-structured clinical interview. Common treatment for depression would use serotonin and/or norepinephrine reuptake inhibitors would worsen symptoms of isolated apathetic patients.

In our nationwide database, we did not routinely re-check the symptoms of apathy and cannot arrange complete neuropsychological examinations to evaluate the symptoms of depression and apathy.

 2. Up to 20% of patients with Parkinson's disease de novo may have mild cognitive impairment. In that sense, it should be clear that these patients were excluded.

ANS: Thanks for your comments. 

Mild cognitive impairment is not uncommon in PD patients. With the restriction of insurance regulations, our clinicians would not be able to routinely arrange yearly neuropsychological examinations for de Novo PD patients except for patients with complaints of subjective memory impairment or other signs or symptoms of cognitive decline. It’s possible for us to exclude PD patients with mild cognitive impairment without regular neuropsychological examinations. However, we exclude PD patients with prior diagnosis of dementia to avoid the inclusion of atypical parkinsonism like Lewy body disease. We have revised our articles in the description of limitations. See them on the Lines 15, 16 of Page 17 of the Discussion section.

 3. Patients with Parkinson's disease and subjective cognitive complaints may have executive dysfunction, which could increase suspicion of depression/apathy. How did the researchers control for this confounding factor?

ANS: Thanks for your comments.

Executive dysfunction, part of mild cognitive impairment, is common in PD patients and the symptoms may be improved after dopaminergic medications. With the restriction of insurance regulations, our clinicians/researches would not be able to routinely arrange yearly neuropsychological examinations for de Novo PD patients except for patients with complaints of subjective memory impairment or other signs or symptoms of cognitive decline. It’s possible for us not being able to exclude PD patients with mild cognitive impairment. However, we exclude PD patients with prior diagnosis of dementia to avoid the inclusion of patients with dementia and atypical parkinsonism like Lewy body disease. We have revised our articles in the description of limitations. See them on the Lines 15, 16 of Page 17 of the Discussion section.

 4. It is important to comment on the likelihood of patients with low levels of education, as in that case, cognitive tests should report on their adaptation and validation.

ANS: Thanks for your comments.

 In our demographic data, we have shown that our PD patients were from different geographic regions of Taiwan with difference economic levels. There is no significant difference between the cases and control. With the restriction of insurance regulations, we would not be able to gain their educational levels and to routinely arrange neuropsychological examinations for patients every year, except for patients with complaints of subjective memory impairment or other signs or symptoms of cognitive decline.

Reviewer #3: Manuscript Number: PONE-D-20-09123. 

Title: Risk factors for depression in patients with Parkinson’s disease: A nationwide nested case-control study. 

The authors report a nested case-control study for risk factors of depression in Parkinson’s disease among 324 patients with depression and 972 patients without depression, using the 2000–2010 Taiwan National Health Insurance Research Database. They found an odds ratio of 1.53 (95% confidence interval, 1.16–2.02; P = 0.003) for anxiety disorders after adjusting for the confounding factors of age, sex, index year, geographic region, urban level, monthly income, and other coexisting medical conditions. For sleep disturbances, the odds ratio was 1.49 (95% confidence interval, 1.14–1.96; P = 0.004) compared to the controls, after adjusting these confounding factors. I enjoyed reading this interesting manuscript. I have only a few suggestions. 

Major points. 

 1. Various kinds of depression. Authors presented ICD codes for various kinds of depression. However, these codes could not be so apparent for readers to describe various aspects of depression in Parkinson’s disease. We can consider the following two systems as a diagnostic nomenclature of depression in Parkinson’s disease.

 1.1. One, the Diagnostic and Statistical Manual of Mental Disorders (5th Edition) by American Psychiatric Association classifies depression according to its etiology and symptom severity. This system comprises of major depression, minor depression, dysthymia, adjustment disorder, and depressive disorder due to Parkinson’s disease.

ANS: Thank you very much for this key point.

 We believed that the Diagnostic and Statistical Manual of Mental Disorders (5th Edition) by American Psychiatric Association can help psychiatrists and neurologist for further clarifying the etiology and symptom severity of depression, particularly in PD with depression. However, we had the limitations in this study because we used the 2000-2010 database, diagnosed using ICD-9-CM system. In this study, we can only divide depressions into “296.2-296.3--major depressive affective disorders”, “atypical depressive disorder”, “300.4--Dysthymic disorder”, and “311--depressive disorder, not elsewhere classified”. See them in S1_Table in detail.

 1.2. The other, the provisional diagnostic criteria for depression in Parkinson's disease proposed by the NINDS/NIMH Work Group classifies depression according to the presence or absence of symptoms and severity. This system includes major depression, minor depression, dysthymia, and subsyndromal depression.

 1.3. I would like to ask authors for supplementing these systems in addition to the ICD codes. If not possible, I’d like to suggest that this aspect could be pointed out as a limitation of this study.

ANS: Thank you for supplying for a simple and clear diagnostic tool to clarifying the etiology and symptom severity of depression in Parkinson’s disease. With the restriction of insurance regulations, we only added this information and a new reference in the limitation of the Discussion section. See it in detail on the Lines 3-8 of Page 18 in the Discussion section and on the Lines 1-5 of Page 28 in the Reference section.

2. Risk factors versus association factors.

If anxiety disorder and sleep disorder are risk factors to develop depression, we can observe a temporal order of preceding risk factors and resulting depression. If not, anxiety disorder and sleep disorder could be ascribed as association factors. Please clarify this point.

ANS: Thanks for key point of risk and association factors.

 In this study, we favored that anxiety disorder and sleep disorder are risk factors by time-order. Anxiety and sleep disturbances were diagnosed before the index date of PD diagnosis and depression diagnosis (in depressive PD group) using both ICD codes and ATC codes in order to confirm the diagnosis and the diagnosis should be made by neurologists or psychiatrists with regular outpatient follow-ups. (See it in detail on the Lines 10-13, Page 9 of the Materials and Methods section.) With some limitations of our study, we think that anxiety disorder and sleep disturbances may be recognized as risk factors for PD patients who develop depression after initial diagnosis.

Minor points. 

 2. Page 35. Figure 1.

In the box of “Excluded due to”, ICD 300.3.

In the box of “PD patients with an incidence of”, ICD 300.4. Please check 300.3. 

ANS: Thanks for your comments.

 We have revised ICD 300.3 to 300.4 in the Figure 1

---

## [Decision Letter · Decision Letter 1]

8 Jul 2020

Risk factors for depression in patients with Parkinson’s disease: A nationwide nested case-control study

PONE-D-20-09123R1

Dear Dr. Tsai,

We’re pleased to inform you that your manuscript has been judged scientifically suitable for publication and will be formally accepted for publication once it meets all outstanding technical requirements.

Kind regards,

Mathias Toft, MD, PhD

Academic Editor

PLOS ONE

Additional Editor Comments (optional):

Reviewers' comments:

Reviewer's Responses to Questions

**Comments to the Author**

1. If the authors have adequately addressed your comments raised in a previous round of review and you feel that this manuscript is now acceptable for publication, you may indicate that here to bypass the “Comments to the Author” section, enter your conflict of interest statement in the “Confidential to Editor” section, and submit your "Accept" recommendation.

Reviewer #2: (No Response)

Reviewer #3: All comments have been addressed

2. Is the manuscript technically sound, and do the data support the conclusions?

Reviewer #2: Yes

Reviewer #3: Yes

3. Has the statistical analysis been performed appropriately and rigorously? 

Reviewer #2: Yes

Reviewer #3: Yes

4. Have the authors made all data underlying the findings in their manuscript fully available?

Reviewer #2: Yes

Reviewer #3: Yes

5. Is the manuscript presented in an intelligible fashion and written in standard English?

Reviewer #2: Yes

Reviewer #3: Yes

6. Review Comments to the Author

Reviewer #2: Congratulations on resolving the observations which have improved the nature of the publication for the benefit of readers.

Reviewer #3: Manuscript Number: PONE-D-20-09123.

Title: Risk factors for depression in patients with Parkinson’s disease: A nationwide nested case-control study.

Thank you very much for sending a revised manuscript. The authors have made a substantial revision of their manuscript and adequately responded to my suggestions. Thank you again for your valuable manuscript.

7. PLOS authors have the option to publish the peer review history of their article (what does this mean?). If published, this will include your full peer review and any attached files.

Reviewer #2: **Yes: **Nilton Custodio

Reviewer #3: **Yes: **Jong-Min Kim

---

## [Editor Report · Acceptance letter]

13 Jul 2020

PONE-D-20-09123R1 

Risk factors for depression in patients with Parkinson’s disease: A nationwide nested case-control study 

Dear Dr. Tsai:

I'm pleased to inform you that your manuscript has been deemed suitable for publication in PLOS ONE. Congratulations! Your manuscript is now with our production department. 

Kind regards, 

on behalf of

Dr Mathias Toft 

Academic Editor

PLOS ONE